# Genomics of Shrimp Allergens and Beyond

**DOI:** 10.3390/genes14122145

**Published:** 2023-11-27

**Authors:** Shanshan Li, Ka Hou Chu, Christine Yee Yan Wai

**Affiliations:** 1School of Life Sciences, The Chinese University of Hong Kong, Hong Kong, China; sshan_li@126.com (S.L.); kahouchu@cuhk.edu.hk (K.H.C.); 2Southern Marine Science and Engineering Guangdong Laboratory (Guangzhou), Guangzhou 510000, China; 3Department of Paediatrics, Prince of Wales Hospital, The Chinese University of Hong Kong, Hong Kong, China; 4Hong Kong Hub of Paediatric Excellence, The Chinese University of Hong Kong, Hong Kong, China

**Keywords:** shellfish allergen, allergen profile, shrimp genomics, cross-reactivity

## Abstract

Allergy to shellfishes, including mollusks and crustaceans, is a growing health concern worldwide. Crustacean shellfish is one of the “Big Eight” allergens designated by the U.S. Food and Drug Administration and is the major cause of food-induced anaphylaxis. Shrimp is one of the most consumed crustaceans triggering immunoglobulin E (IgE)-mediated allergic reactions. Over the past decades, the allergen repertoire of shrimp has been unveiled based on conventional immunodetection methods. With the availability of genomic data for penaeid shrimp and other technological advancements like transcriptomic approaches, new shrimp allergens have been identified and directed new insights into their expression levels, cross-reactivity, and functional impact. In this review paper, we summarize the current knowledge on shrimp allergens, as well as allergens from other crustaceans and mollusks. Specific emphasis is put on the genomic information of the shrimp allergens, their protein characteristics, and cross-reactivity among shrimp and other organisms.

## 1. Introduction

Food allergy refers to the adverse immune responses triggered by the consumption or exposure to proteins present in various types of food, predominantly resulting from immunoglobulin E (IgE)-mediated Type I hypersensitivity reactions. Up to 10% of the world’s population suffers from food allergies [1,2,3]. Shellfish, a term commonly used in fisheries and colloquial speech, refers to aquatic invertebrates possessing exoskeletons and harvested for sustenance. This diverse category encompasses mollusks, crustaceans, and echinoderms. Though the majority are gathered from saline environments, certain varieties inhabit limnic habitats as well. Shellfish allergy specifically involves an allergic reaction to proteins present in shellfish, including some crustaceans in the order Decapoda (e.g., shrimp, lobster, and crab) and mollusks from the class Bivalvia (e.g., clam, mussel, and oyster). The prevalence of shellfish allergy ranges from 0% to 10.3%, depending on the method of diagnosis and population (usually more prevalent in Asia) [4,5,6] and is increasing in both developed and developing countries [7,8,9,10]. In some instances, exposure to cooking vapor containing shellfish allergens can also provoke an allergic reaction [11]. Apart from being one of the “Big Eight” allergens designated by the U.S. Food and Drug Administration, shellfish is also the top food item leading to admission to an emergency department [12]. 

Shellfish allergies can manifest in a wide range of symptoms, ranging from mild to severe and, in some cases, life-threatening. Common symptoms include skin reactions (e.g., hives, itching, swelling), gastrointestinal distress (e.g., nausea, vomiting, diarrhea), respiratory symptoms (e.g., coughing, wheezing, shortness of breath), and in severe cases, anaphylaxis, a severe allergic reaction that can cause difficulty in breathing, a drop in blood pressure, and loss of consciousness [13]. When an individual with a shellfish allergy consumes or comes into contact with specific shellfish proteins, their immune system recognizes these proteins as foreign and initiates an allergic reaction. The exact reasons why allergies occur are not fully understood, but several factors contribute to the development of food allergies, including genetic predisposition [13,14,15,16] and environmental factors [14,17]. While most children outgrow food allergies for eggs and milk with age, an allergy to shellfish typically lasts a lifetime [18]. 

There are four primary groups of shrimps that are commonly recognized. These include the suborder Dendrobranchiata and the infraorders Procarididea, Stenopodidea, and Caridea. Shrimp, in particular the penaeid shrimp (family Penaeidae), are the most consumed crustacean food item, and shrimp allergy is among the most common shellfish allergies, especially in regions where shrimp consumption is widespread [7,19,20]. Knowing the specific allergens responsible for triggering food allergies is paramount for effectively managing allergies. This aids in developing advanced diagnostic tools, more targeted therapies, and potential preventive strategies, such as immunotherapy or allergen-specific treatments. At present, there are a total of ten shrimp allergens registered with the World Health Organization and International Union of Immunological Societies (WHO/IUIS) Allergen Nomenclature Database (Table 1). Among the registered allergens, nine were identified from penaeid shrimps, including greasyback shrimp *Metapenaeus ensis* (De Haan, 1844), brown shrimp *Penaeus (Farfantepenaeus) aztecus* (Ives, 1891)*,* Indian prawn *Penaeus (Fenneropenaeus) indicus* (H. Milne Edwards, 1837)*,* western king prawn *Penaeus (Melicertus) latisulcatus* (Kishinouye, 1896), black tiger shrimp *Penaeus monodon* (Fabricius, 1798), and whiteleg shrimp *Penaeus (Litopenaeus) vannamei* (Boone, 1931). Six of the shrimp allergens were identified in caridean shrimp (infraorder Caridea), including North Sea shrimp *Crangon. crangon* (Linnaeus, 1758), Siberian prawn *Exopalaemon modestus* (Heller, 1862), giant freshwater prawn *Macrobrachium rosenbergii* (De Man, 1879), and northern shrimp *Pandalus borealis* (Krøyer, 1838). 

## 2. Shrimp Allergens Identified by Immunodetection

Conventionally, the identification of novel allergens relies on immunodetection and mainly immunoblotting. In these experimental studies, shrimp proteins are extracted from the muscle and screened using sera from shellfish-allergic subjects for detecting IgE-binding proteins. These proteins are subsequently identified with mass spectrometry for their peptide fingerprints, followed by matching to protein databases, such as NCBI and UniProt. For instance, our group recently identified ten IgE-binding proteins from the shrimp *P. monodon* based on immunodetection and comprehended a shrimp allergen panel comprising 11 recombinant shrimp allergens [37]. These include tropomyosin (TM, Pen m 1), arginine kinase (AK, Pen m 2), myosin light chain (MLC, Pen m 3), sarcoplasmic calcium-binding protein (SCP, Pen m 4), troponin C (TnC, Pen m 6), hemocyanin (Hc, Pen m 7), triosephosphate isomerase (TIM, Pen m 8), fatty-acid-binding protein (FABP, Pen m 13), and glycogen phosphorylase (GP, Pen m 14), plus two potential allergens, enolase (Eno) and aldolase (ALDA).

## 3. Shrimp Allergens Identified by the Transcriptomic Approach

With the advances in sequencing technologies and bioinformatics analysis, the transcriptomic approach has been adopted to identify shrimp allergens and characterize potential cross-reactivity [39]. In a novel study, Karnaneedi et al. [40] uncovered and compared the complete transcriptome of shrimp and identified 39 potential novel shrimp allergens from five species. The authors conducted a de novo assembly and analysis of the transcriptomes from five commonly consumed penaeid shrimp species, including *P. vannamei*, *P. monodon*, banana shrimp *Penaeus (Fenneropenaeus) merguiensis* (De Man, 1888), *P. latisulcatus*, and endeavor shrimp *Metapenaeus endeavouri* (Schmitt, 1926), and established an in-house reference allergen database by collecting the amino acid sequences of 2172 allergens from two allergen databases (WHO/IUIS Allergen Nomenclature Database and Food Allergy Research and Resource Program (FARRP) Allergen Protein Database). A pairwise identity cut-off value of 50% was then used to indicate potential allergenicity and assess the probability of a shrimp protein being an allergen. As a result, BLAST analysis revealed a significant number of matches to allergen sequences, primarily associated with shellfish, mites, and fungi, resulting in the identification of 40 allergen sequences from *P. vannamei*, 44 from *P. monodon*, 42 from *P. merguiensis*, 44 from *P. latisulcatus*, and 50 from *M. endeavouri*. In total, seven previously identified crustacean allergens were confirmed among the five shrimp species (TM, AK, SCP, MLC, TnC, TnI, TIM), and the variances in the abundance of each allergen within individual shrimp species were investigated. On the other hand, other unreported potential allergens were identified, including heat shock protein 70 (HSP 70), α-tubulin, chymotrypsin, β-enolase, Eno, aldolase A, glyceraldehyde-3-phosphate dehydrogenase (GAPDH), and cyclophilin (CyPs), which are possibly responsible for clinical cross-reactivity, such as among crustaceans, mites, and insects. This research provides evidence that the transcriptomic approach offers advantages in discovering and comparing the whole repertoire of shellfish allergens at high resolution in addition to putative novel allergens.

## 4. Genomic and Proteomic Characteristics of Shrimp Allergens

Based on the WHO/IUIS allergen registry, the most comprehensive allergen profile was delineated from *P. monodon*, comprising nine registered shrimp allergens, including TM, AK, MLC2, SCP, Tn, Hc, TIM, FABP, and GP. These allergens are mostly identified by conventional immunodetection methods as mentioned. Yet the availability of the genome assembly data of penaeid shrimp has revolutionized our understandings on gene regulation, including information on chromosome locations, repetitive elements, alternative splicing, translational regulation, and methylation. Such knowledge is crucial not only for unraveling the intricate mechanisms underlying gene expression and its impact on biological processes but also has led to new allergen discoveries in penaeid shrimp.

### 4.1. Genome Assembly of Penaeus

Among shrimp, the genome assembly data for four *Penaeus* species, namely, *P. chinensis* (Osbeck, 1765), *P. japonicus* (Spence Bate, 1888), *P. monodon*, and *P. vannamei*, along with annotation information, are available in the NCBI (National Center for Biotechnology Information) database (Table 2). A high-quality genome assembly of *P. indicus* was also reported, but unfortunately, the annotation information for this species is currently unavailable. 

Zhang et al. [41] presented the first high-quality genome in penaeid shrimp, on the most commercially important species, *P. vannamei*. The genome sequence of *P. vannamei* spans approximately 1.66 gigabases (Gb) with a scaffold N50 of 605.56 kilobases (kb) and contains 25,596 protein-coding genes. The genome exhibits a high proportion of simple sequence repeats (>23.93%). The expansion of genes related to vision and locomotion suggests adaptations to a benthic environment. The intensified ecdysone signal pathway, facilitated by gene expansion and positive selection, may explain the frequent molting observed in penaeid shrimp.

Researchers utilized Illumina and Oxford Nanopore Technologies platforms to generate a draft genome assembly of *P. japonicus* [42]. The assembly spanned 1.70 Gb with 18,210 scaffolds and had a scaffold N50 of 234.9 kb, exhibiting 34.38% GC content. The genome displayed a high proportion of simple repeats (27.4%) and included 26,381 predicted protein-coding gene models, with functional annotations available for 68.2% of the genes. 

The whole genome of *P. monodon* was reported based on a chromosome assembly [43]. The researchers successfully assembled a high-quality genome sequence by utilizing a combination of long-read sequencing technologies including Pacific Biosciences (PacBio), Chicago, and Hi-C. The final assembly covered 92.3% of the estimated genome size, totaling 2.39 Gb, with a scaffold N50 of 44.9 megabases (Mb), and consisted of 44 pseudomolecules, reflecting the haploid chromosome number. A significant portion of the assembly (62.5%) was composed of repetitive elements, the highest reported among crustacean species. 

Katneni et al. [44] presented a high-quality genome assembly of *P. indicus*, which spans 1.93 Gb with a scaffold N50 of 34.4 Mb and contains 28,720 protein-coding genes and 49.31% repeat elements. Notably, the *P. indicus* assembly exhibits the highest proportion of simple sequence repeats (31.99%) among sequenced animal genomes and demonstrates a superior sequence contiguity compared to other shrimp genomes. The assembly also provides valuable resources, including 15,563 coding single nucleotide polymorphisms (SNPs), for genetic improvement programs, evolutionary studies, and stock management in penaeid shrimp fisheries and culture.

A chromosome-level genome assembly of *P. chinensis* revealed significant genomic adaptations [45]. The assembled genome, with a size of 1.47 Gb, including 57.73% repetitive sequences, anchors to 43 pseudochromosomes, with a scaffold N50 of 36.87 Mb. In total, 25,026 protein-coding genes were predicted. It displays contraction compared to other penaeid species, potentially due to migration. The genome also exhibits expanded gene families associated with cellular and metabolic processes, as well as contracted gene families linked to virus infection processes, indicating the species adaptation to migration and cold environments. Additionally, the analysis identified genes associated with metabolism, phototransduction, and the nervous system in cultured shrimps, suggesting targeted artificial selection during domestication and providing valuable insights for understanding genetic changes during evolution.

The availability of whole genome sequence assemblies in the above species allows for the identification of genetic markers associated with allergen genes in *Penaeus* shrimp. These markers can be used to predict the presence or absence of allergen genes in individuals, enabling the selection of breeding candidates with reduced allergenicity. This approach can contribute to the development of hypoallergenic shrimp varieties, enhancing the safety and acceptance of shrimp products for allergic individuals.

**Table 2 genes-14-02145-t002:** Gene information of allergens characterized in *Penaeus* shrimps.

Species	Genome Assembly	References	Gene ID and Location ^1^									
			Tropomyosin	Arginine Kinase	Myosin Light Chain 2	Sarcoplasmic Calcium-Binding Protein	Myosin Light Chain 1	Troponin C	Hemocyanin	Triosephosphate Isomerase	Fatty-Acid-Binding Protein	Glycogen Phosphorylase-like Protein
*P. chinensis*	size: 1.47 Gb; contig N50: 472.84 kb; scaffolds:1060; scaffold N50: 36.9 Mb; repetitive sequences: 57.73%.	[45]	ID: 125048035; Chromosome 42, NC_061860.1 (2872728..2923947)	ID: 125046934; Chromosome 39, NC_061857.1 (23292279..23322424)	ID: 125032710; Chromosome 2, NC_061820.1 (36562377..36563543)	ID: 125043149; Chromosome 33, NC_061851.1 (2573366..2588962, complement)	ID: 125040575; Chromosome 29, NC_061847.1 (27675878..27684529, complement)	ID: 125028981; Chromosome 1, NC_061819.1 (25531203..25539418)	ID: 125046232; Chromosome 38, NC_061856.1 (21846290..21853098)	ID: 125028271; Chromosome 8, NC_061826.1 (13786682..13788417, complement)	ID: 125029003; Chromosome 9, NC_061827.1 (10506608..10523531)	ID: 125029446; Chromosome 10, NC_061828.1 (34855913..34890537, complement)
*P. japonicus*	size: 1.70 Gb; contig N50: 132.8 kb; scaffolds: 18,210; scaffold N50: 234.9 kb; simple sequence repeats: 27.4%.	[42]	ID: 122251322; NW_025030369.1 (4708..55143)	ID: 122249885; NW_025030159.1 (44864..71320, complement)	ID: 122266134; NW_025035186.1 (37625..38804)	ID: 122264462; NW_025034254.1 (89818..105301)	ID: 122259789; NW_025032380.1 (8332..16949, complement)	ID: 122262516; NW_025033339.1 (96181..103665, complement)	ID: 122243277; NW_025037757.1 (41553..44524)	ID: 122250573; NW_025030266.1 (162108..163787)	ID: 122257811; NW_025031762.1 (205217..217336, complement)	ID: 122246447; NW_025029730.1 (398826..415764, complement)
*P. monodon*	size: 2.39 Gb; contig N50: 45.2 kb; scaffolds: 26634; scaffold N50: 44.9 Mb; repetitive elements: 62.5%.	[43]	ID: 119594951; Chromosome 35, NC_051420.1 (921718..971934	ID: 119591270; Chromosome 28, NC_051413.1 (1599636..1601570, complement)	ID: 119570076; Chromosome 4, NC_051389.1 (10051929..10053917, complement)	ID: 119585264; Chromosome 19, NC_051404.1 (46480220..46495804)	ID: 119587760; Chromosome 23, NC_051408.1 (7267583..7276463)	ID: 119578966; Chromosome 11, NC_051396.1 (43393497..43401039)	ID: 119590770; Chromosome 27, NC_051412.1 (30008096..30012410, complement)	ID: 119572953; Chromosome 5, NC_051390.1 (9171969..9173836)	ID: 119574570; Chromosome 6, NC_051391.1 (45560875..45573117, complement)	ID: 119580811; Chromosome 14, NC_051399.1 (6470562..6490308, complement)
*P. vannamei*	size: 1.66 Gb; contig N50: 86.9 kb; scaffolds: 4682 kb; scaffold N50: 605.6 kb; simple sequence repeats: >23.93%	[41]	ID: 113820940; NW_020870691.1 (252085..301985)	ID: 113816366; NW_020870178.1 (42354..45892)	ID: 113816291; NW_020870168.1 (222691..223847)	ID:113814611; NW_020870007.1 (194583..205932)	ID: 113822686; NW_020870901.1 (724534..733535, complement)	ID: 113828663; NW_020872422.1 (8101..17716)	ID: 113823617 NW_020871007.1 (22864..26603, complement)	ID: 113802550; NW_020872930.1 (498965..500845)	ID: 113815521; NW_020870091.1 (1715084..1730731)	ID: 113800223; NW_020872700.1 (140879..160091, complement)

**^1^** Gene ID (GenBank) and location are retrieved from NCBI (National Center for Biotechnology Information).

### 4.2. Tropomyosin

TM belongs to a family of muscle proteins and has been identified as the primary allergenic component responsible for triggering allergic reactions in individuals with shrimp allergies. In 1981, Hoffman et al. [46] initially characterized TM as an allergen in *P. aztecus* (Pen a 1). Subsequently, it was also identified as an allergen in *P. monodon* (Pen m 1) by Shanti et al. in 1993, and in *M. ensis* (Met e 1) by Leung et al. in 1994 [24]. Subsequently, TM was identified as an allergen in other shrimp species, including other penaeid shrimps, such as *P. vannamei* and *P. indicus*, as well as caridean shrimps, including *C. crangon*, *E. modestus*, giant freshwater prawn *M. rosenbergii*, and northern shrimp *P.s borealis* (Table 1). TM accounts for allergic reactions in at least 80% of individuals allergic to shrimp, as it binds approximately 80% of the shrimp-specific IgE in these subjects. In shrimp, the coding DNA sequence (CDS) size of TM is 855 bp, while the gene length differs among different species (Table 1). The location of the TM gene in *P. monodon* is Chromosome 35, NC_051420.1 (921718..971934, complement), with 24 exons. At the protein level, the length of TM is approximately 284 amino acid residues(AA), with a molecular weight (MW) of 34–38 kDa. The structural stability and resistance to heat and digestion of TM contribute to its allergenicity [47,48]. Efforts have been made to mitigate or diminish the allergenicity of shrimp, minimize potential health risks through different processing methods, and provide strategies for the immunotherapy of shrimp allergy. Various food processing technologies have been shown to have the potential to reduce the allergenic properties of shrimp TM, including ultrasound-assisted high temperature–pressure [48], high-methylglyoxal during thermal processing [49], glycation modification [50,51,52,53,54], peroxidation product modification [55,56], and dietary polyphenol treatment [57]. 

### 4.3. Arginine Kinase

After tropomyosin, AK has been identified as crustaceans’ second most important allergen. It shows positive IgE binding in 10–51% of individuals with shrimp allergies [58]. AK is an enzyme that plays a crucial role in cellular energy metabolism and provides a rapid and localized energy source for muscle contraction. In 2003, Yu et al. [59] first identified arginine kinase as a new allergen (Pen m 2) in *P. monodon* with all six selected shrimp-allergic patients reacting positive with natural Pen m 2 in a skin test. Similar results were subsequently demonstrated in other shrimp species, including *C. crangon*, *P. vannamei*, and *M. rosenbergii* (Table 1). In shrimp, the CDS size of AK in *C. crangon*, *P. monodon*, and *P. vannamei* is 1071 bp, and in *M. rosenbergii*, 1068 bp, while the gene length differs among different species (Table 1). The location of the AK gene in *P. monodon* is Chromosome 28, NC_051413.1 (1599636..1601570, complement), with two exons. At the protein level, the length of AK is approximately 356 AA, with a MW of 40–45 kDa. AK is unstable under thermal processing and easier to degrade in acidic conditions than in alkaline conditions [60]. Moreover, AK is relatively stable at 20–40 °C and begins to unfold and lose its secondary structure at 55–70 °C, followed by the cleavage of disulfide bonds at 70–80 °C and aggregate formation at 90–100 °C. During pH processing, acidic conditions (pH ≤ 5) resulted in more damage to the secondary structure [61]. Yet AK remains a significant food allergen despite its unstable physicochemical properties. Mei et al. [62] modified the conformational structure and epitopes of AK from the mud crab *Scylla paramamosain* (Estampador, 1950) by site-directed mutagenesis. Fei et al. [63] reported that the enzymatic cross-linking of AK using tyrosinase and caffeic acid, followed by thermal polymerization, shows promising potential in reducing its IgE-binding activity and allergenicity. This process involves modifying the molecular and immunological characteristics of the allergen. 

### 4.4. Myosin Light Chain

MLC is a component of the myosin protein complex, which is responsible for muscle contraction. There are two types of myosin light chains: the essential light chain (MLC1) and the regulatory light chain (MLC2). MLC2 was first identified as a shrimp allergen in *P. vannamei* (Lit v 3) in 2008 [33]; in this study, immunoblotting demonstrated IgE binding by 21/38 (55%) serum samples with recombinant MLC. While tropomyosin is recognized as the most prevalent allergen in crustaceans, it is noteworthy that some patients exhibited predominant binding to Lit v 3. In two patients, Lit v 3 was the sole allergen recognized. This indicates that the inclusion of Lit v 3 in future diagnostic and therapeutic strategies holds significant importance. Later, MLC2 was identified as an allergen in *P. monodon* (Pen m 3), and MLC1 was identified as an allergen in *C. crangon* (Cra c 5) (Table 1). The CDS size of Lit v 3 and Pen m 3 is 534 bp (MLC 2), and the size of Cra c 5 (MLC1) is 462 bp. The gene length differed in different species (Table 1). The location of the MLC2 gene in *P. monodon* is Chromosome 4, NC_051389.1 (10051929..10053917), with one exon. At the protein level, the length of MLC is approximately 153 AA, with a MW of 17–20 kDa. MLC stayed stable when exposed to different temperatures, even up to 100 °C. Also, its allergenicity did not change much between 30 and 100 °C [64], and it remained stable at various pH levels, both acidic and alkaline. As a minor allergen, the abundance of MLC is substantially lower than other primary allergens in muscle.

### 4.5. Sarcoplasmic Calcium-Binding Protein

SCP is a protein in muscle cells that plays a crucial role in calcium regulation and muscle contraction. SCP has been recognized as a significant allergen in our recent study on *P. monodon*, comparable in importance to TM [37]. Approximately 29% to 50% of individuals with shrimp allergies exhibit positive IgE binding to SCP, with an even higher frequency of 59% observed among children. However, unlike TM, which has been extensively studied, research on SCP remains relatively limited. SCP was identified as an allergen in *C. crangon* (Cra c 4), with 3/8 (38%) shrimp-allergic patients having IgE binding to recombinant Cra c 4 in immunoblotting and 11/31 (35%) patients having positive binding on ImmunoCAP [21]. It was named Lit v 4 by Ayuso et al., with 31/52 (60%) pediatric shellfish-allergic subjects recognizing SCP in the boiled *P. vannamei* extract. In *P. monodon*, it was named Pen m 4 with 8 of 16 crustacean-allergic sera reacting to natural Pen m 4 by fluorescence ELISA [35]. Our study showed that SCP has a sensitization rate of 28% by ELISA with recombinant Pen m 4 [37]. The CDS size and gene length of shrimp SCP is 582 bp, and the location of the Pen m 4 gene is Chromosome 19, NC_051404.1 (46480220..46495804), with eight exons (Table 1). At the protein level, the protein length of SCP is approximately 192 AA, with a MW of 20–25 kDa. It has been reported that SCP exhibits high resistance to acid–alkali conditions and heat [65]. Zhao et al. [66] examined the immunological properties and structural changes of the recombinant Lit v 4 (rLit v 4) under various temperature conditions. They found that rLit v 4 exhibits a distribution of secondary structures as follows: 60.62% α-helix, 4.15% β-sheet, 12.95% β-turn, and 22.28% random coil. Moreover, rLit v 4 exhibited stable IgE-binding reactivity up to 80 °C, but higher thermal processing led to a significant decline in the capacity to bind IgG/IgE, accompanied by changes in both secondary and tertiary structures. 

### 4.6. Troponin 

Troponin is a complex of three proteins, troponin C (TnC, Ca^2+^-binding subunit), troponin I (TnI, tropomyosin-binding subunit), and troponin T (TnT, which inhibits the interaction between actin and myosin T), that play a crucial role in regulating muscle contraction. Among these subunits, TnC and Tn I exhibit IgE reactivity. TnC is a less common crustacean allergen, with an allergic sensitization rate around 20%. This is lower than the sensitization rates of TM, AK, or SCP. It was identified as an allergen in *C. crangon* (Cra c 6), with 9 of 31 shrimp-allergic sera having positive binding on ImmunoCAP [21]. In *P. monodon*, 8 of 35 shrimp-allergic individuals had IgE that reacted to TnC (Pen m 6) in immunoblot and ELISA [36] (Table 1). The CDS size and gene length of shrimp TnC is 453 bp, and the location of the Pen m 6 gene is Chromosome 19, NC_051396.1 (43393497..43401039), with six exons (Table 1). At the protein level, the length of Cra c 6 and Pen m 6 is 150 AA, with a MW of 16.8–21 kDa. TnC has been reported as a heat-stable allergen in the Asian green mussel *Perna viridis* (Linnaeus, 1758) as it retained IgE reactivity in the immunoblots of extracts from cooked mussels [67]. 

### 4.7. Hemocyanin

Hc is a copper-containing metalloprotein that is an oxygen carrier in the blood (hemolymph) of various invertebrates. They are typically present in the hemolymph rather than enclosed in blood cells like hemoglobin in the red blood cells of vertebrates. Hc molecules can be large, composed of hexamers or multi-hexamers, and often consisting of multiple subunits with a MW of ~75 kDa [68]. As the circulatory tissues are not always removed during food preparation, there may be high concentrations of Hc present in cooked shrimp. Mendoza et al. [68] verified the presence of at least 12 distinct Hc isoforms in shrimp hemolymph and confirmed putative Hc gene assemblies using transcriptomic data. These findings facilitate the observation of specific Hc isoform expression in shrimp hemolymph under various environmental, nutritional, and pathogenic conditions. Hc was first identified as a heat-stable allergen in *M. rosenbergii* by Piboonpocanun et al. in 2011 [69,70] and was also identified as an allergen in Lanchester’s freshwater prawn *M. lanchesteri* (De Man, 1911) [70]. Our group recognized Hc as an allergen in *P. monodon* (Pen m 7), with seven of 32 subjects with DBPCFC-confirmed shrimp allergy showing IgE binding by ELISA to a recombinant Hc expressed in insect cells [37] (Table 1). The CDS size and gene length of Pen m 7 is 2052 bp, and the location of the gene is Chromosome 19, NC_051396.1 (43393497..43401039), with six exons (Table 1). At the protein level, the length of Hc in shrimp is usually 662–683 AA, with a MW of around 76 kDa. Guillen et al. [71] found that Hc in *P. vannamei* seems to lose its allergenicity at high temperatures (heating shrimp extract at 60 °C for 10 min), as no IgE binding was observed in the heated extract. However, Piboonpocanun et al. [50] indicated that dissociated or monomeric forms of Hc were not precipitated and not degraded when boiled. Following their investigation using SDS gel and immunoblot analysis of dialyzed and boiled hemolymph from both *P. monodon* and *M. rosenbergii*, the authors concluded that boiling does not cause the degradation or impairment of IgE binding ability in the monomeric form of Hc.

### 4.8. Triosephosphate isomerase

TIM is an essential enzyme in glycolysis, a metabolic pathway that breaks down glucose to produce energy in the form of ATP. TIM is found in nearly all living organisms, from bacteria to humans, and its structure and function are highly conserved across species. TIM was characterized as a crustacean allergen for the first time in *C. crangon* (Cra c 8) with 5/8 (63%) shrimp-allergic patients showing IgE binding to Cra c 8 in immunoblotting, and 7/31 (23%) shrimp-allergic sera having positive binding to Cra c 8 on ImmunoCAP [21]. TIM was later identified as Pen m 8 in *P. monodon* with 12/30 (40%) of *P. monodon*-allergic subjects showing positive IgE binding to this allergen on ELISA, and 2/12 (17%) subjects being positive in a basophil activation test (Table 1). The CDS size and gene length of Cra c 8 is 750 bp, while the 1466 bp long Pen m 8 has a CDS of 800 bp. Pen m 8 is located at Chromosome 5, NC_051390.1 (9171969..9173836), with four exons (Table 1). At the protein level, the length of TIM in shrimp is usually 249–288 AA, with a MW of 27–28 kDa. Utilizing far-ultraviolet CD spectra, it was determined that native TIM contains 31.7% α-helices, 12.4% antiparallel extended strands, and 7.4% parallel extended strands [72]. It was also demonstrated by dot-blot analysis that the secondary structure was notably affected during heat treatment, particularly at 100 °C, and the IgE-binding activity of TIM decreased as the temperature exceeded 60 °C. In addition, extreme acidic conditions (pH 1.0) or alkaline conditions (pH 11.0) resulted in a reduction in α-helices in the structure. Notably, the IgE-binding activity of TIM remained relatively stable under acidic and alkaline conditions. Intriguingly, an increase in IgE-binding activity was observed at pH 2–3. 

### 4.9. Fatty-Acid-Binding Protein

FABPs are a family of small, conserved proteins involved in the intracellular transport and metabolism of fatty acids. As shown in Table 1, it was reported as an allergen in *P. vannamei* (Lit v 13) with 10/36 (28%) shrimp-allergic subjects having IgE binding to recombinant Lit v 13 by ELISA [38]. In *P. monodon* (Pen m 13), 20/30 (67%) allergic subjects reacted positively on ELISA, and 7/12 (58%) showed positive basophil activity to the recombinant allergen [37]. Yet FABP has not yet been recognized as an allergen in other shellfish species. The CDS size of Lit v 13 and Pen m 13 is 411 bp, while the gene length differs slightly (Table 1). The FABP gene in *P. monodon* is located at Chromosome 6, NC_051391.1 (45560875..45573117, complement), with four exons. At the protein level, the length of FABP in shrimp is 246 AA, with a MW of 15–20 kDa. The heat and pH stability of this allergen is unknown. 

### 4.10. Glycogen Phosphorylase

GP is an enzyme that plays a key role in the breakdown of glycogen, a branched polymer of glucose that serves as the primary energy storage molecule in animals. The heat stability of this allergen is unknown. The sole study reporting GP as an allergen was conducted by our group [37], with 8/17 (47%) oral-food-challenged confirmed shrimp-allergic subjects reacting positively on ELISA against recombinant GP, while 3/17 (18%) also showed positive basophil reactivity upon GP stimulation. The CDS size of the GP gene is 2559 bp, with a total gene length of 2701 bp (Table 1). The GP gene of *P. monodon* is located on Chromosome 14, NC_051399.1 (6470562..6490308, complement), with seven exons. At the protein level, the length of GP in shrimp is around 852 AA, with a MW of about 95 kDa. The heat and pH stability of this allergen is unknown.

## 5. Other Potential Shrimp Allergens

Besides the allergens discussed above, several other proteins present in shrimp have also been reported as potential allergens. A study showed that eleven (68%) and seven (43%) patients demonstrated IgE-binding activity to titin (Ttn) (identified by mass spectrometry) in raw and heated *P. monodon* extracts, respectively [29]. Phosphopyruvate hydratase/enolase in *P. monodon* was identified as a novel, putative shrimp allergen but with a limited number of patients and pediatric patients involved only [73]. Khanaruksombat et al. [74] also identified enolase as a potential important allergen in the muscle of *P. merguiensis*, accompanied with the potential allergen myosin heavy chain (MHC). Moreover, GAPDH showed allergenic reactions in the muscle and shell of *P. merguiensis*, and vitellogenin (VG) exhibited a high intensity in immunoblot analysis across all vitellogenic stages which indicated it as an important allergen in the ovaries of *P. merguiensis*. Additionally, the authors suggested ovarian peritrophin 1 precursor (SOPs), β-actin, and 14-3-3 protein as novel but minor allergens in *P. merguiensis*. Similarly, the purification of the protein is needed to confirm its allergenicity. Our group also demonstrated the IgE-binding ability of enolase and aldolase from *P. monodon* by immunoblot, but their recognition frequency was low, while the recombinant protein also showed low sensitization rates [37]. Pyruvate kinase in *P. vannamei* showed a high specific IgE binding in raw and cooked shrimp extracts in seven (100%) and four (57%) of seven allergic patients’ sera, respectively [75]. The registration of these allergens has been partly hindered by the limited sample size and/or the absence of protein purification necessary for confirming their allergenicity. Despite this, the identified proteins hold potential diagnostic and therapeutic values for studies on shrimp allergies.

## 6. Cross-Reactivity of Shrimp Allergens

Patients with shrimp allergy often exhibit allergic symptoms to other crustaceans and mollusks, as well as IgE cross-reactivity with nonedible arthropods such as insects (cockroaches) and arachnids (mites) due to the highly conserved allergens. Among all, TM contributed to the majority of cross-reactivity detected. Due to its involvement in essential biological processes across various organisms, TM is a widely distributed invertebrate pan-allergen with highly conserved sequences and structures [47,76,77], and thus it exhibits strong cross-reactivity. Studies have demonstrated that TM is an important allergen in other crustaceans, such as crabs (Cha f 1, Por p 1, Scy p 1) and lobsters (Pan s 1, Hom a 1, Pan s 1), as well as mollusks such as oysters (Cra a 1, Cra g 1, Sac g 1), gastropods (Hal l 1, Hal m 1, Hel as 1), and squid (Tod p 1) (Table 3). In addition to shellfish, it is characterized as a food allergen in herring worm (Ani s 3) [78], common roundworm (Asc l 3), silk moth (Bomb m 3) [79], and *Mozambique tilapia* (Ore m 4) [80], as an airway allergen in mosquitoes (Aed a 10) [81], cockroaches (Bla g 7, Per a 7) [82,83], mites (Blo t 10, Cho a 10, Der f 10, Der p 10, Lep d 10, Tyr p 10) [84,85,86,87,88,89], and termites (Copt f 7) [81], and as an injection allergen in midges (Chi k 10) [90]. AK is also a widely distributed invertebrate pan-allergen with remarkably conserved sequences and shows cross-reactivity with allergens from crab (Cal b 2, Scy p 2) and oyster (Cra a 2) (Table 3). Moreover, it has been identified as a food allergen in moths (Cal b 2, Plo i 1) [91,92] and as an airway allergen in cockroaches (Bla g 9) [93,94] and mites (Der f 20, Der p 20, Per a 9, Tyr p 20) [95,96,97]. MLC shows cross-reactivity among crustacean shellfish, viz. crab (Scy p 3), lobster (Hom a 3), and crayfish (Pro c 5) (Table 3). Interestingly, it was also identified as a food allergen in vertebrates including chicken (Gal d 7) [98] and cattle (Bos d 13) [99], and as an airway allergen in cockroaches (Bla g 8, Per a 8) [100] and mites (Der f 26, Der p 26) [39]. Additionally, the cross-reactivity of SCP was investigated through sequence and immunoblotting analyses by Zhao et al. [66]. They demonstrated that crustacean SCP showed high sequence identities ranging from 77% to 96%. The sequence homology of crustacean SCPs with those of insects, mites (and their relative, the horseshoe crab), and mollusks was generally lower, ranging from 14% to 55% [65]. Thus far, SCP shows cross-reactivity in crab (Scy p 4), oyster (Cra a 4), and lobster (Pon l 4) (Table 3). It was also identified as an airway allergen in mosquito (Aed a 5) [81] and cattle (Bos d 3) [101]. TnC was identified as an allergen in American lobster (Hom a 6) recognized by 24% of the patients. On the other hand, TnI has been identified in narrow-clawed crayfish *Pontastacus leptodactylus* (Eschscholtz, 1823) as 2/25 (8%) of shrimp-allergic patients had IgE that reacted with Pon l 7 in IgE immunoblotting [102]. TnC has also been identified as an airway allergen in cockroaches (Bla g 6, Per a 6) [103] and mites (Der f 39, Der p 39, Tyr p 34) [104,105]. It was shown that 5/87 (5.75%) of house-dust-mite patients reacted with Der p 39, and 5/47 (10.6%) of the tested patients showed IgE binding with Tyr p 34. Hc was found to exhibit IgE activity in crabs, such as purple mud crab *Scylla tranquebarica* (Fabricius, 1798) [106] and Chinese mitten crab *Eriocheir sinensis* (H. Milne Edwards, 1853) [106,107]. Additionally, it was identified as an airway allergen in German cockroach (*Blattella germanica*) (Bla g 3) [108,109] and American cockroach *Periplaneta americana* (Linnaeus, 1758) (Per a 3) [110]. TIM was found as a cross-reactivity allergen in other crustaceans such as crab (Scy p 8) [111], lobster (Arc s 8), and crayfish (Pro c 8) [89] (Table 3), as well as a food allergen in catfish (Pan h 8) [112]. Additionally, TIM has been reported as an airway allergen in wheat (Tri a 31) [113], mold (Asp t 36) [114], tree pollen extract (Pla a 7), and mite (Der f 25, Der p 25) [115], and as an injection or inhalation allergen in mosquito (Aed a 12). In addition to shrimp, FABP was first recognized as an allergen in storage mite *Blomia tropicalis* (Bronswijk, Cock and Oshima, 1973) (Blo t 13) in 1997 [116], followed by other mites (Aca s 13, Lep d 1, Tyr p 13, Der f 13, Der p 13) [117,118,119,120].

## 7. Allergens from Other Crustaceans and Mollusks 

On top of the cross-reactive allergens, there are novel allergens identified in crustaceans and mollusks that were not reported in shrimp. For instance, filamin C was identified as a crab allergen named Scy p 9, with 30/100 shellfish-allergic subjects showing IgE binding to natural and recombinant Scy p 9 as shown by Western blot, dot blot, and ELISA [121]. Yu et al. [122] found a 99 kDa protein paramyosin in whelk *Rapana venosa* (Valenciennes, 1846) that displayed specific IgE binding with sera from sea-snail-allergic patients, which was then identified as a novel allergen named Rap v 11. Unlike filamin C, paramyosin has also been identified as an allergen in herring worm (Ani s 2) [123] and mites (Blo t 11, Der f 11, Der p 11, Tyr p 11) [124,125,126].

## 8. Conclusions

To date, ten shrimp allergens have been registered in the WHO/IUIS Allergen Nomenclature Database (TM, AK, MLC2, SCP, MLC1, TnC, Hc, TIM, FABP, GP), and several potential shrimp allergens have been reported. Cross-reactivity among different shellfish species is widespread and encompasses components that extend beyond the major allergen TM. Except for FABP and GP, most of the allergens show clinical cross-reactivity not only between crustaceans and mollusks but also with other arthropods like mites and insects, and in some cases, even vertebrates and plants. During the last two decades, significant progress has been made in identifying and characterizing shrimp allergens and beyond. 

The availability of whole genome sequence assemblies of penaeid shrimp species, including *P. chinensis*, *P. indicus*, *P. japonicus*, *P. monodon*, and *P. vannamei* provides an opportunity to reveal the genomic information of allergen genes in these species and the influence of genomic features on transcriptional regulation, alternative splicing, and the allergenicity of the proteins. With this comprehensive genomic information, the potential for genomic selection in shrimp breeding programs can be further explored.

The advancement in sequencing technologies also improves the delineation of shellfish allergens at high resolution compared to conventional immunodetection methods. In the era of precision medicine and based on studies showing the heterogenous allergen profile of shrimp- and shellfish-allergic patients, such a comprehensive panel allows precision diagnosis using the component-resolved diagnosis approach. Recombinant technologies, on the other hand, facilitate the modification of allergens to mitigate their allergenicity and the recombinant fusion of allergens to suitable carriers for vaccine construction. In summary, with our better understanding of shrimp allergens along with their genomic information, the achievement of precision diagnosis and treatment of shellfish allergy will just be a matter of time.

## Figures and Tables

**Table 1 genes-14-02145-t001:** Allergens characterized in shrimp.aq.

Biochemical Identity ^1^	MW (kDa) ^2^	Heat Stability	Route of Exposure	Allergen Nature and Physiological Function	Species	Allergen Identity IUIS Name	GenBank Protein Accession No.	Protein Size (aa) ^3^	Sensitization Rate (IgE Binding)	References
Tropomyosin	34–38	highly heat stable and IgE reactive	ingestion inhalation	Muscle contraction coiled-coil protein that binds to actin and regulates the interaction of troponin and myosin	*Crangon. crangon* (North Sea shrimp)	Cra c 1	ACR43473	284	11/25 (44%) in immunoblot	[21]
					*Exopalaemon modestus* (Siberian prawn)	Exo m 1	__	__	18/18 (100%) in immunoblot and ELISA	[22]
					*Macrobrachium rosenbergii* (Giant freshwater prawn)	Mac r 1	ADC55380	284	10/13 (78%) in ELISA	[23]
					*Metapenaeus ensis* (Greasyback shrimp)	Met e 1	AAA60330	274	__	[24]
					*Pandalus borealis* (Northern shrimp)	Pan b 1	CBY17558	284	7/8 (88%) in skin-prick test 4/6 (67%) in positive basophil activation 5/5 (100%) in immunoblot. 4/4 (100%) in ELISA	[25]
					*Penaeus aztecus* (Brown shrimp)	Pen a 1	AAZ76743	284	28/34 (82%) in skin-prick test	[26]
					*Penaeus indicus* (Indian prawn)	Pen i 1	__	__	__	[27]
					*Penaeus latisulcatus* (Western king prawn)	Mel l 1	AGF86397	284	10/18 (56%) in immunoblot	[28]
					*Penaeus monodon* (Black tiger shrimp)	Pen m 1	AAX37288	284	11/16 (69%) in immunoblot	[29]
					*Penaeus vannamei* (Whiteleg shrimp)	Lit v 1	ACB38288	284	15/15 (100%) in peptide microarray analysis	[30]
Arginine kinase	40–45	labile but can elicit IgE-binding	ingestion inhalation	Enzyme catalyzes the reversible transfer of the phosphoryl group from ATP to arginine	*C. crangon* (North Sea shrimp)	Cra c 2	ACR43474	356	3/8 (38%) in immunoblot. 9/31 (29%) in ImmunoCAP	[21]
					*M. rosenbergii* (Giant freshwater prawn)	Mac r 2	ADN88091	355	8/48 (18%) in skin-prick test 37/48 (77%) in basophil hexosaminidase test	[31]
					*P. monodon* (Black tiger shrimp)	Pen m 2	AAO15713	356	5/18 (27%) in immunoblot	[29]
					*P. vannamei* (Whiteleg shrimp)	Lit v 2	ABI98020	356	__	[32]
Myosin light chain 2	17–20	stable	ingestion	Regulatory function in smooth muscle contraction when phosphorylated by MLC kinase	*P. monodon* (Black tiger shrimp)	Pen m 3	ADV17342	177	7/10 (70%) in immunoblot and ELISA	[29]
					*P. vannamei* (Whiteleg shrimp)	Lit v 3	ACC76803	177	17/19 (90%) in immunoblot	[33]
Sarcoplasmic calcium-binding protein	20–25	stable	ingestion	Binds to cytosolic calcium (Ca^2+^) and acts as a calcium buffer regulating calcium-based signaling	*C. crangon* (North Sea shrimp)	Cra c 4	ACR43475	193	3/8 (38%) in immunoblot 11/31 (35%) in ImmunoCAP	[21]
					*P. monodon* (Black tiger shrimp)	Pen m 4	ADV17343	193	8/16 (50%) in ELISA	[34]
					*P. vannamei* (Whiteleg shrimp)	Lit v 4	ACM89179	193	31/52 (60%) in immunoblot	[35]
Myosin light chain 1	17.5 kDa	stable	ingestion	Regulatory function in smooth muscle contraction when phosphorylated by MLC kinase	*C. crangon* (North Sea shrimp)	Cra c 5	ACR43477	153	5/8 (63%) in immunoblot 6/31 (19%) in ImmunoCAP	[21]
Troponin C	16.8–21	stable	ingestion	Regulates the interaction of actin and myosin during muscle contraction on binding to calcium	*C. crangon* (North Sea shrimp)	Cra c 6	CR43478	150	4/8 (50%) in immunoblot 9/31 (29%) in ImmunoCAP	[21]
					*P. monodon* (Black tiger shrimp)	Pen m 6	ADV17344	150	8/35 (23%) in immunoblot and ELISA	[36]
Hemocyanin	76	stable	ingestion	Transports oxygen throughout the body	*P. monodon* (Black tiger shrimp)	Pen m 7	AEB77775	683	3/17 (18%) in ELISA	[37]
Triosephosphate isomerase	27–28	labile	ingestion inhalation	Key enzyme in glycolysis; catalyzes the conversion of dihydroxyacetone phosphate to glyceraldehyde 3-phosphate	*C. crangon* (North Sea shrimp)	Cra c 8	ACR43476	249	5/8 (63%) in immunoblot 7/31 (23%) in ImmunoCAP	[21]
					*P. monodon* (Black tiger shrimp)	Pen m 8	ADG86240	266	12/30 (40%) in ELISA 2/12 (17%) in basophil activation test	[37]
Cytoplasmic fatty-acid-binding protein	15–20	__	ingestion	Facilitates the transfer of fatty acids between extra- and intracellular membranes	*P. monodon* (Black tiger shrimp)	Pen m 13	AEP84100	136	20/30 (67%) in ELISA 7/12 (58%) in basophil activation test	[37]
					*P. vannamei* (Whiteleg shrimp)	Lit v 13	ADK66280	136	10/36 (28%) in ELISA	[38]
Glycogen phosphorylase-like protein	95	__	ingestion	Enzymes catalyze the rate-limiting step in glycogenolysis in animals	*P. monodon* (Black tiger shrimp)	Pen m 14	URW11955	852	8/17 (47%) in ELISA 3/17 (18%) in basophil activation	[37]

^1^ For allergen proteins, see www.allergen.org. ^2^ MW, molecular weight. ^3^ aa, amino acid. — = data deficient

**Table 3 genes-14-02145-t003:** List of characterized shellfish allergens.

			Shellfish Species	Tropomyosin *	Arginine Kinase *	Myosin Light Chain 2 *	Sarcoplasmic Calcium-Binding Protein *	Myosin Light Chain 1 *	Troponin C, Troponin I *	Hemocyanin *	Triosephosphate Isomerase *	Filamin C *	Paramyosin *	Cytoplasmic Fatty-Acid-Binding Protein *	Glycogen Phosphorylase-like Protein *
Crustaceans	Shrimp	Penaeid shrimp	*Metapenaeus ensis* (Greasyback shrimp)	Met e 1											
			*Penaeus aztecus* (Brown shrimp)	Pen a 1											
			*Penaeus indicus* (Indian prawn)	Pen i 1											
			*Penaeus latisulcatus* (Western king prawn)	Mel l 1											
			*Penaeus monodon* (Black tiger shrimp)	Pen m 1	Pen m 2	Pen m 3	Pen m 4		Pen m 6	Pen m 7	Pen m 8			Pen m 13	Pen m 14
			*Penaeus vannamei* (Whiteleg shrimp)	Lit v 1	Lit v 2	Lit v 3	Lit v 4								
		Caridean shrimp	*Crangon. crangon* (North Sea shrimp)	Cra c 1	Cra c 2		Cra c 4	Cra c 5	Cra c 6		Cra c 8				
			*Exopalaemon modestus* (Siberian prawn)	Pen a 1											
			*Macrobrachium rosenbergii* (Giant freshwater prawn)	Mac r 1	Mac r 2										
			*Pandalus borealis* (Northern shrimp)	Pan b 1											
	Crab		*Callinectes bellicosus* (Warrior swimming brown crab)		Cal b 2										
			*Charybdis feriatus* (Crucifix crab)	Cha f 1											
			*Eriocheir sinensis* (Chinese mitten crab)												
			*Portunus pelagicus* (Blue swimmer crab)	Por p 1											
			*Scylla paramamosain* (Mud crab)	Scy p 1	Scy p 2	Scy p 3	Scy p 4				Scy p 8	Scy p 9			
	Lobster		*Homarus americanus* (American lobster)	Hom a 1		Hom a 3			Hom a 6						
			*Panulirus stimpsonii* (Spiny lobster)	Pan s 1											
			*Archaeopotamobius sibiriensis*								Arc s 8				
	Crayfish		*Pontastacus leptodactylus* (Narrow-clawed crayfish)				Pon l 4		Pon l 7						
			*Procambarus clarkii* (Red swamp crayfish)	Pro c 1	Pro c 2			Pro c 5			Pro c 8				
	Brine shrimp		*Artemia franciscana* (San Francisco brine shrimp)					Art fr 5							
Mollusks	Gastropod		*Haliotis laevigata x Haliotis rubra* (Jade tiger abalone)	Hal l 1											
			*Haliotis midae* (Perlemoen abalone)	Hal m 1											
			*Helix aspersa [Cornu aspersum]* (Brown garden snail)	Hel as 1											
			*Rapana venosa* (Veined *rapa* whelk)										Rap v 11		
	Bivalve		*Crassostrea angulata* (Portuguese oyster)	Cra a 1	Cra a 2		Cra a 4								
			*Crassostrea gigas* (Pacific oyster)	Cra g 1											
			*Saccostrea glomerata* (Sydney rock oyster)	Sac g 1											
	Cephalopod		*Todarodes pacificus* (Japanese flying squid)	Tod p 1											

* Allergens stated are registered with the WHO/IUIS Allergen Nomenclature.

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
