# Peer review of "Genomics of Shrimp Allergens and Beyond"

_genes, 2023, doi:10.3390/genes14122145_

Round 1

Reviewer 1 Report

Comments and Suggestions for Authors

The manuscript concerns allergens identified mainly in shrimps, but also in other invertebrates. The article is well written and I have no meritorical based objections. However, I believe that it should provide more detailed information on how allergens affect patients (medical symptoms, responding organs), both in adults and children.

Shrimps are very popular food component for people worldwide. Their aquaculture production increases. The MS concerns mainly allergens in shrimps. The authors intended to summarize the present knowledge on the issues of genomic characteristics of main groups of known allergens. They recommend genomics as more efficient method of their discovery in comparison with immunological methods.  

Ms. compiles and summarizes the current knowledge about allergens based on transcriptomics and emphasizes that a more comprehensive use of genomics will open new possibilities for counteracting allergies by, for example, constructing vaccines.  

There is no a review paper published so far, with such approach to allergens in shrimps and focused on this important sea food component. Nevertheless, I suggest updating the literature cited, published in 2022-2023.   

The manuscript is well written, the text is logically organized possible to understand for none-expert in the field. The conclusions result from the data compiled and quoted and are related to the main question.  

Author Response

  • Comment 1: It should provide more detailed information on how allergens affect patients (medical symptoms, responding organs), both in adults and children.

Response: Thank you for this suggestion. We have provided some information about allergy symptoms, and responding organs in the introduction part (on page 1, paragraph 2, lines 42-49).

  • Comment 2: Updating the literature cited, published in 2022-2023.

Response: Thank you for pointing this out. We respond by adding several latest literature on page 16.

Reviewer 2 Report

Comments and Suggestions for Authors

Dear Editor,

The review paper entitled “Shrimp Allergens, and Beyond” is appropriately well-written, structured and developed by LI et al. in suitable English with a clear structure. They reviewed recent advances in allergic compounds in shrimp, crustaceans and mollusks. They investigated the allergic proteins and the associated genes. This study is interesting; however, there are some main points of view which should be addressed and clarified before making a final decision.

1.      My main concern is the correspondence and association of the journal and the study. Obviously, genes encoding allergen-related proteins have been discussed in the paper; but this aspect of the allergic compounds which are related to the aim and scope of the journals should be more observable in the main sections of the manuscript such as the abstract and the title of the manuscript.

2.      Keywords are not chosen appropriately. Please revise and develop them in the correct forms.

3.      Please summarize the conclusions section and point out only the main conclusions in this section. 

Author Response

  • Comment 1: Contents related to the aim and scope of the journals should be more observable in the main sections of the manuscript such as the abstract and the title of the manuscript.

Response: Thank you for this comment. we have modified both the title (Genomics of Shrimp Allergens and Beyond) and the abstract.

  • Comment 2: Keywords are not chosen appropriately.

Response: Thank you for this suggestion. we have modified the keywords, “shellfish allergen; allergen profile; shrimp genomics, cross-reactivity

  • Comment 3: Please summarize the conclusions section and point out only the main conclusions in this section.

Response: Thank you for this suggestion. we have revised the conclusion section accordingly on page 16, line 488-490: “In summary, with our better understanding on the shrimp allergens along with their genomic information, the achievement of precision diagnosis and treatment of shellfish allergy would be just a matter of time”.

Reviewer 3 Report

Comments and Suggestions for Authors

The manuscript is a useful and comprehensive review on shrimp allergens. However, the following minor comments need to be addressed before it can be accepted for publication:

-Delete the comma in the title. Furthermore, it should be defined to which specific taxon you refer with the trade name “shrimp”

-Abbreviation IgE (Immunoglobulin E) needs to be explained upon first mentioning in the Abstract and Introduction

-Abstract: Use simple present (“is”) instead of “will be” and write “put” instead of “focused”

-Introduction: as mentioned for the title, imprecise taxon names need to be defined (e.g. shrimp, lobster, crab, clam, mussel, oyster). Furthermore, the term shellfish needs to be defined as it can be used both broadly and specifically (either referring to any exoskeleton-bearing aquatic invertebrates used as food, e.g. anything from clams to lobster, or only filter-feeding mollusks to the exclusion of crustaceans and other invertebrates. And although the term is particularly applied to marine spp., the term sometimes includes edible freshwater invertebrates.

-Lines 58 etc.: Species authorities should be added upon first mentioning of a species name throughout the manuscript. Depending on the journal, this might be done without indicating the year of the original description (usually the reference is not cited in the list of references then) or with indication of the year of the original description, with the reference to be included in or excluded from the list of references.

-The manuscript should be checked for typos and moderate language issues throughout. One example is line 122, where it must be “the” instead of “The”. Another example is the double parenthesis in the last line of Table 2.

-Abbreviation Kb (kilobases; first mentioning in line 129) needs to be explained. Also, consistently write “kb” instead of “Kb”.

-Also explain other abbreviations upon first mentioning, such as Mb (megabases) and Gb (gigabases).

Comments on the Quality of English Language

Only minor modifications needed as explained in my review.

Author Response

  • Comment 1: Delete the comma in the title. Furthermore, it should be defined to which specific taxon you refer with the trade name “shrimp”.

Response: Thank you for your suggestions. We have modified the title (Genomics of Shrimp Allergens and Beyond). For the definition of “shrimp”, we added information on page 2, lines 55-57, “There are four primary groups of shrimp that are commonly recognized. These include the suborder Dendrobranchiata and the infraorders Procarididea, Stenopodidea, and Caridea.”

  • Comment 2: Abbreviation IgE (Immunoglobulin E) needs to be explained upon first mentioning in the Abstract and Introduction.

Response: Thank you for this suggestion. we have included its full name on page 1, lines 15 and 28.

  • Comment 3: Abstract: Use simple present (“is”) instead of “will be” and write “put” instead of “focused”.

Response: Thank you for this suggestion. We have revised accordingly on page 1, line 21.

  • Comment 4: Introduction: as mentioned for the title, imprecise taxon names need to be defined (e.g. shrimp, lobster, crab, clam, mussel, oyster). Furthermore, the term shellfish needs to be defined as it can be used both broadly and specifically (either referring to any exoskeleton-bearing aquatic invertebrates used as food, e.g. anything from clams to lobster, or only filter-feeding mollusks to the exclusion of crustaceans and other invertebrates. And although the term is particularly applied to marine spp., the term sometimes includes edible freshwater invertebrates.

Response: Thank you for this comment. We have added an explanation accordingly on page 1, lines 29-35. “Shellfish, a term commonly used in fisheries and colloquial speech, refers to aquatic invertebrates possessing exoskeletons and harvested for sustenance. This diverse category encompasses mollusks, crustaceans and echinoderms. Though the majority are gathered from saline environments, certain varieties inhabit limnic habitats as well. Shellfish allergy specifically involves an allergic reaction to proteins present in shellfish, including some crustaceans in order Decapoda (e.g., shrimp, lobster, crab) and mollusks from class Bivalvia (e.g., clam, mussel and oyster).”

  • Comment 5: -Lines 58 etc.: Species authorities should be added upon first mentioning of a species name throughout the manuscript. Depending on the journal, this might be done without indicating the year of the original description (usually the reference is not cited in the list of references then) or with indication of the year of the original description, with the reference to be included in or excluded from the list of references.

Response: Thank you for this suggestion. We have added species authorities in lines 66-74, 100, 101, 134, 237, 301, 317,436, 441, 442, 450, 451, 460.

  • Comment 6: The manuscript should be checked for typos and moderate language issues throughout. One example is line 122, where it must be “the” instead of “The”. Another example is the double parenthesis in the last line of Table 2.

Response: Thank you for your comments. We apologize for the typos and have revised the manuscript accordingly.

  • Comment 7: Abbreviation Kb (kilobases; first mentioning in line 129) needs to be explained. Also, consistently write “kb” instead of “Kb”. Also explain other abbreviations upon first mentioning, such as Mb (megabases) and Gb (gigabases).

Response: Thank you for your comments. We have revised accordingly in lines 141 and 157.

Reviewer 4 Report

Comments and Suggestions for Authors

Dear Authors,

your review work about allergenic molecules of shrimp, in particular, is well organized and detailed. The in depth-analysis of transcriptomic approach is of great interest. I only suggest to implement the review with a paragrapfh dedicated to a major clarification of potentiality of this approach.

Furthermore, if you consider it appropriate and relevant, some few information about immunological and molecular assays, available for food safety/food fraud and cross-contamination purposes in shrimp field, could be interesting.

I suggest the re-formatting of Table 3, maybe changing sheet orientation.

Please check the format of references list in according to "Genes" guidelines.

Best regards.

Comments on the Quality of English Language

Minor editing of English language required

Author Response

  • Comment 1: Suggest to implement the review with a paragraph dedicated to a major clarification of potentiality of this approach.

Response: Thank you for this suggestion. we have summarized the conclusion section and added a paragraph (page16, lines 488-490, please see above).

  • Comment 2: Add some few information about immunological and molecular assays, available for food safety/food fraud and cross-contamination purposes in shrimp field, could be interesting.

Response: Thank you for this suggestion. It would have been interesting to include this aspect, however, in the case of this literature review, it seems slightly out of our subjects’ focus on allergen profile. Therefore, information about immunological and molecular assays for food safety, food fraud, and cross-contamination purposes in the shrimp field was not included in the revised manuscript.

  • Comment 3: I suggest the re-formatting of Table 3, maybe changing sheet orientation.

Response: Thank you for this suggestion. We would like the production office to help reorganize the table if necessary to fit into the journal format.

  • Comment 4: [Please check the format of references list in according to "Genes" guidelines.]

Response: Thank you for the comments. We have checked the reference formats.

Round 2

Reviewer 2 Report

Comments and Suggestions for Authors

Dear authors,

Thank you for your responses and revising.

I have no more comments.